# Platinum retention in plasma, urine, and normal colonic mucosa in cisplatin-treated testicular cancer survivors

Emilie C. H. Breekveldt[1,2], Berbel L. M. Ykema[1,3], Alwin D. R. Huitema[4,5,6], Jourik A. Gietema[7], Jos H. Beijnen[4], Petur Snaebjornsson[8,9], Michael Schaapveld[10], Flora E. van Leeuwen[10], Hilde Rosing[4], Monique E. van Leerdam[1,3] *, on behalf of the CATCHER study working group[¶]

1 Department of Gastrointestinal Oncology, Netherlands Cancer Institute, Amsterdam, The Netherlands,
2 Department of Public Health, Erasmus University Medical Center, Rotterdam, The Netherlands,
3 Department of Gastroenterology and Hepatology, Leiden University Medical Center, Leiden, The Netherlands, 4 Department of Pharmacy and Pharmacology, Netherlands Cancer Institute, Amsterdam, The Netherlands, 5 Department of Pharmacology, Princess Máxima Center for Pediatric Oncology, Utrecht, The Netherlands, 6 Department of Clinical Pharmacy, University Medical Center Utrecht, Utrecht University, Utrecht, The Netherlands, 7 Department of Medical Oncology, University Medical Center Groningen, Groningen, The Netherlands, 8 Department of Pathology, Netherlands Cancer Institute, Amsterdam, The Netherlands, 9 Faculty of Medicine, University of Iceland, Reykjavik, Iceland, 10 Department of Epidemiology, Netherlands Cancer Institute, Amsterdam, The Netherlands

¶ The complete list of members of the CATCHER study working group can be found in the Acknowledgments section.
* M.E.van_Leerdam@lumc.nl

**Data Availability Statement:** The data are owned by the Netherlands Cancer Institute, and we are not allowed to share them. Data are available upon

## Abstract

Testicular cancer survivors (TCS) treated with platinum-based chemotherapy have increased cancer risk. Platinum retention in healthy tissue may contribute to carcinogenesis. We assessed total platinum concentrations in plasma, urine, and normal colonic mucosa samples in TCS treated with cisplatin. From the total TCS treated with ≥3 cycles cisplatin who participated in a colonoscopy-screening study in four Dutch hospitals (n = 154), plasma (n = 131) and urine (n = 115) samples were collected. During colonoscopy, 60 biopsies of normal colonic mucosa (two samples each per 30 randomly selected patients undergoing colonoscopy) were obtained. Samples were analyzed for total platinum concentrations using inductively coupled plasma mass spectrometry and compared with controls (plasma: 10, urine: 3, normal colonic mucosa: 9). The median age at colonoscopy was 50 years (interquartile range (IQR): 43–57) and the median time since treatment was 20 years (IQR:16–26). Median platinum concentrations in plasma (38 pg/mL; IQR: 24–61 pg/mL) and urine (376 pg/mL; IQR: 208–698 pg/mL) remained elevated in TCS up to 40 years post-treatment and were higher than in controls (all controls were below limits of detection [plasma: 25 pg/mL, urine: 6 pg/mL]). The median platinum concentration in normal colonic mucosa was 0.58 pg/mg (IQR: 0.33–1.59 pg/mg) in the transverse and 0.51 pg/mg (IQR:0.26–1.25 pg/mg) in the descending colon. Cisplatin treatment is associated with long-term retention of platinum in various patient sample types. This might increase cancer risk by causing somatic mutations, potentially explaining the elevated risk of second malignant

reasonable request through the Clinical Trial Service Unit from the Netherlands Cancer Institute [trialbureau@nki.nl].

**Funding:** The author(s) received no specific funding for this work.

**Competing interests:** P. Snaebjornsson has done unrelated consultancy for MSD and Bayer and received payment from MEDtalks for educational presentation. All other authors declare no conflicts of interest. These companies had no involvement in study design; collection, management, analysis and interpretation of data; or the decision to submit for publication.

neoplasms in TCS. The long-term effects of platinum retention should be monitored to understand carcinogenesis and to provide guidelines for early second cancer detection.

Trial registration: ClinicalTrials.Gov: NCT04180033.

## Introduction

Cisplatin is widely used in treatment of various malignancies, such as ovarian, bladder, head-and-neck, esophageal, breast, brain and lung cancer. Cisplatin is also essential in the systemic treatment of testicular cancer (TC), typically consisting of bleomycin, etoposide/ifosfamide, and cisplatin [1]. The use of cisplatin has resulted in remarkably high 5-year overall survival rates around 90%, depending on the stage at diagnosis [1, 2]. Despite its efficacy, cisplatin is associated with several adverse effects, including nephrotoxicity, ototoxicity, cardiotoxicity and neurotoxicity. There is accumulating evidence that in some cases, prior anti-cancer treatment is associated with the development of second malignant neoplasms (SMNs) [3]. Treatment with cisplatin-based chemotherapy has been associated with increased risk of developing gastrointestinal (GI) and other SMNs in TCS [4]. Uptake of cisplatin into cells occurs both through passive diffusion as well as various modes of transport. Within the cell, cisplatin subsequently induces DNA damage by multiple mechanisms of action, both directly by forming DNA cross-links and indirectly through multifaceted cellular damage. Depending on the response, the cell may survive or undergo apoptosis [5]. While most cisplatin will covalently bind to proteins and is cleared by the kidneys, a small amount accumulates in rapidly growing tissues, both tumor tissue as well as proliferating healthy tissue [5]. The retention and accumulation of cisplatin in healthy tissues may exert long-term carcinogenic effects and may ultimately lead to the formation of SMNs.

Platinum has previously been demonstrated in plasma and urine of TCS treated with (cis) platinum-based chemotherapy even more than a decade after treatment [6, 7]. Other platinum-based agents, such as oxaliplatin, have also been measured in human tissues, albeit for shorter periods after treatment and at lower concentrations [8]. The causation of late side-effects by platinum-based chemotherapy is complex, and understanding the contribution of long-term retention of cisplatin to carcinogenesis of SMNs might be crucial for effective prevention or early detection of SMNs in TCS. This study aims to investigate whether platinum is still detectable in plasma, urine, and normal colonic mucosa of TCS up to 40 years after cisplatin-based chemotherapy. In addition, we assessed the correlation between platinum concentrations in urine and plasma and platinum concentrations in normal colonic mucosa.

## Methods

### Participants, collected samples and samples analyzed

All samples were retrieved from participants in the CATCHER study, a colonoscopy screening study in four hospitals in the Netherlands, which aimed to evaluate the diagnostic yield of colonoscopy in TCS treated with platinum-based chemotherapy. The CATCHER study design has been described previously [9]. All participants met the following criteria: 1) TC diagnosis before age 50, 2) TC treatment consisted of at least 3 cycles of cisplatin-containing chemotherapy, 3) TC treatment was at least 8 years ago, 4) age at enrollment ≥35 and ≤75 years, 5) detection of colorectal neoplasia was considered beneficial taking into account co-morbidities. This study was approved by the Medical Ethical Committee (study number M19CTR, clinical trial

number: NCT04180033) and the institutional review board (study numbers IRB22-083 and IRB22-222) of the Netherlands Cancer Institute. Data and materials were anonymously processed. Patient-derived tissue and data were collected, stored, and used in accordance with the Code of Conduct for the Proper Secondary Use of Human Tissue in the Netherlands, Dutch Federation of Biomedical Scientific Societies, The Netherlands.

A total of 154 individuals provided written informed consent to participate in the CATCHER study. Study enrollment started on February 18th, 2020, and ended on November 25th, 2022. Colonoscopy was performed in 137 individuals. Plasma and urine samples were collected at the enrollment visit or prior to colonoscopy. A total of 131 plasma and 115 urine samples were collected and for 106 individuals, both plasma and urine samples were available. Nine individuals provided a urine sample only and 25 individuals provided a plasma sample only. A random selection of 30 individuals was made from the study participants who underwent colonoscopy. For each patient, one transverse colon biopsy and one descending colon biopsy were used for analyses, for a total of 60 normal colon tissue samples. Biopsies from normal colonic mucosa of the transverse and descending colon were obtained during colonoscopy and neoplasia was removed according to standard protocol. For 29 participants, samples from plasma, urine and normal colonic mucosa were available.

The Institutional Review Board approved the search for (biobanked) control samples consisting of patients treated at the Netherlands Cancer Institute who had never received platinum-based chemotherapy and who were matched for male sex and age to the CATCHER study participants. We obtained 10 plasma, three urine, and nine normal colonic mucosa samples as control samples.

## Outcomes

Primary outcomes were total platinum concentrations in plasma, urine, and normal colonic mucosa samples of the transverse and descending colon. Secondary outcomes were platinum half-lives in plasma and urine, median platinum concentrations by most advanced lesion at colonoscopy, and the association between time since treatment and number of cycles of cisplatin (independent variable) and log-transformed platinum concentrations in plasma, urine and normal colonic mucosa (dependent variable). The most advanced lesion at colonoscopy was categorized into i) no lesions, ii) non-advanced adenomas or non-advanced serrated polyps (SPs), and iii) advanced neoplasia (AN). Advanced neoplasia was defined as either advanced adenomas (AAs), advanced serrated polyps (ASPs), or CRC. AA was defined as any adenoma measuring ≥10 mm and/or having high-grade dysplasia and/or histologically confirmed villous component ≥25%. ASP was defined as at least one SP ≥10 mm, sessile serrated lesion with dysplasia, or traditional serrated adenoma.

## Clinical parameters

Information regarding cumulative cisplatin dose and follow-up time were collected from patient files for all TCS included in the colonoscopy screening study. The standard dose regimen of cisplatin is 20 mg/m$^2$ cisplatin for 5 days for each cycle. TCS received either 3, 4, or >4 cycles of cisplatin during treatment; 3 cycles of cisplatin are defined as <350 mg/m$^2$, 4 cycles as 350–450 mg/m$^2$, and >4 cycles as >450 mg/m$^2$.

## Sample retrieval and measurement of platinum

Inductively coupled plasma mass spectrometer (ICP-MS) was used to quantify the total platinum concentration in plasma, urine and normal colonic mucosa samples. The total platinum concentration refers to all platinum-containing species within the sample, including the intact

cisplatin molecule as well as any platinum metabolites or forms bound to proteins or biomolecules. Sample preparation is described in detail in the *Supplementary Materials*. Pretreated, diluted samples were introduced into the ICP-MS (ICP-MS 7800, Agilent Technologies, Santa Clara, CA, USA) for quantification of total platinum concentrations. Calibration standards and quality control samples were prepared from carboplatin in human plasma [10]. The concentration range of the calibration standards were 50–5000 pg/mL. For the quantification of total platinum concentrations in normal colonic mucosa samples, the concentration range of the calibration standards was 10–1000 pg/mL. To fit the calibration data (response ratio Pt 194/Ir 191 vs. the concentration), linear regression was applied with a weighting factor of $1/x^2$, where x is the total platinum concentration. Quality control (QC) samples were included in each analytical run (at least 6 samples containing platinum at low, medium and high concentration over the calibration range). For every analytical run, the measured platinum concentrations of at least 2/3$^{rd}$ of the QC samples should be within the ±15% deviation from the nominal concentration and at least 50% at each level should meet this criterion. For all executed analytical runs, the acceptance criteria were met.

The platinum concentration in the study samples was quantified if the concentration was measured within the concentration range of the calibration standards. In plasma and normal colonic mucosa samples, however, total platinum concentrations were frequently below the lower limit of quantitation (LLOQ). Therefore, the limit of detection (LOD) was defined as a signal to noise ratio of at least 3 and the concentration of samples between the LLOQ en LOD were semi-quantitatively reported. The lower limit of quantitation (LLOQ) for plasma was set to 50 pg/mL (lowest calibrations standard concentration) and the LOD at 25 pg/mL. For normal colonic mucosa samples, the LLOQ and LOD were 10 pg/mL and 2 pg/mL, respectively.

## Statistical analyses

Descriptive statistics were used to summarize the data, including median and interquartile range (IQR) and 95% confidence intervals (95%CI). Differences in platinum concentrations between groups were analyzed using the Mann-Whitney-U or Kruskal-Wallis test. Associations between platinum concentrations and time since last cisplatin cycle were assessed using scatter plots. Correlations between plasma and urine concentrations of platinum were evaluated by the Spearman correlation coefficient. Plasma and urine platinum half-lives were estimated from single measurements at various time points since treatment of participants in the CATCHER cohort. Linear regression analysis of ln-transformed plasma or urine platinum concentrations (dependent variable) and time since TC treatment (independent variable) was used to approximate platinum half-lives [Model 1]. Observations were excluded when platinum concentrations were below the LOD. Outliers were identified by computing z-scores for each data point. Data points with a z-score exceeding the threshold of 3 were considered outliers and were also excluded. Platinum half-lives were estimated using the following formula: $\frac{-\ln(2)}{coeff(model\ 1)}$. Linear regression analyses were performed to determine the associatlion between number of cycles of cisplatin and time since treatment and log-transformed platinum concentrations in plasma, urine and normal colonic mucosa. Data were analyzed using R version 4.0.2.

## Results

The median age of TCS at colonoscopy was 50 years (IQR: 43–57) and the median time since treatment was 20 years (IQR: 16–26; Table 1). Median age at TC diagnosis was 27.5 (IQR 23–34). Most TCS received three or four cycles of cisplatin. No CRCs were detected during colonoscopy.

**Table 1. Characteristics of the study population.**

| | Age at enrollment (median (y), IQR) | Time since TC treatment (median (y), IQR) | Age at TC diagnosis (median (y), IQR) | Cycles of cisplatin (n, %) |
|---|---|---|---|---|
| Study participants (n = 154) | 50 (43–58) | 20 (16–27) | 28 (23.3–34) | 3: 50 (32.5)<br>4: 85 (55.2)<br>≥5: 16 (10.4)<br>Unknown: 3 (1.9) |
| Participants who underwent colonoscopy (n = 137) | 50 (43–57) | 20 (16–26) | 27.5 (23–34) | 3: 43 (31.4)<br>4: 76 (55.5)<br>≥5: 15 (10.9)<br>Unknown: 3 (2.2) |
| Plasma samples (n = 131) | 50 (43–58) | 20 (16–27) | 28 (23.5–34) | 3: 41 (31.3)<br>4: 72 (55.0)<br>≥5: 15 (11.5)<br>Unknown: 3 (2.2) |
| Urine samples (n = 115) | 50 (32–59) | 21 (17–27.8) | 28 (23–33.5) | 3: 36 (31.3)<br>4: 65 (56.5)<br>≥5: 12 (10.4)<br>Unknown: 2 (1.7) |
| Normal colonic mucosa samples (n = 30) | 49 (40–55) | 17 (15–21) | 29 (24–34) | 3: 11 (36.6)<br>4: 15 (50)<br>≥5: 4 (13.3)<br>Unknown: 0 |
| Plasma + urine samples (n = 106) | 50.5 (43–59) | 20 (16–28) | 28 (24–34) | 3: 32 (30.2)<br>4: 60 (56.6)<br>≥5: 12 (11.3)<br>Unknown: 2 (1.9) |
| Plasma + urine + normal colonic mucosa samples (n = 29) | 49 (40–55) | 17 (15–21) | 29 (24–34) | 3: 11 (37.9)<br>4: 15 (51.7)<br>≥5: 3 (10.3)<br>Unknown: 0 |

## Platinum in plasma

The median platinum concentration in plasma (n = 131) was 38 pg/mL (IQR: 24–61 pg/mL). A total of 34.4% of platinum plasma concentrations in TCS was equal to or above the LLOQ (50 pg/mL), 41.2% was between the LLOQ and the LOD (25 pg/mL); and 24.4% was below the LOD and not used for the calculations (Fig 1A). All platinum concentrations in control samples (n = 10) were below the LOD. The estimated platinum half-life in plasma was 13.3 years (95%CI: 8.9–17.7; Fig 1B). The platinum concentration in plasma significantly decreased with time since treatment (S1 Table in S1 File). The platinum concentration in plasma significantly increased with number of cycles of cisplatin administered at baseline.

## Platinum in urine

The median platinum concentration in urine (n = 115) was 376 pg/mL (IQR: 208–698 pg/mL). Almost all (94.8%) platinum urine concentrations in TCS were above the LLOQ (50 pg/mL), 4.6% was between the LLOQ and the LOD (6 pg/mL), and only one sample was below the LOD and excluded from the dataset (Fig 2A). All platinum concentrations in control samples (n = 3) were below the LOD. The estimated platinum half-life in urine was 9.8 (95%CI: 6.9–12.6) years (Fig 2B). The platinum concentration in urine significantly decreased with time since treatment (S2 Table in S1 File). The highest dose of cisplatin administration at baseline was significantly associated with the platinum concentration in urine.

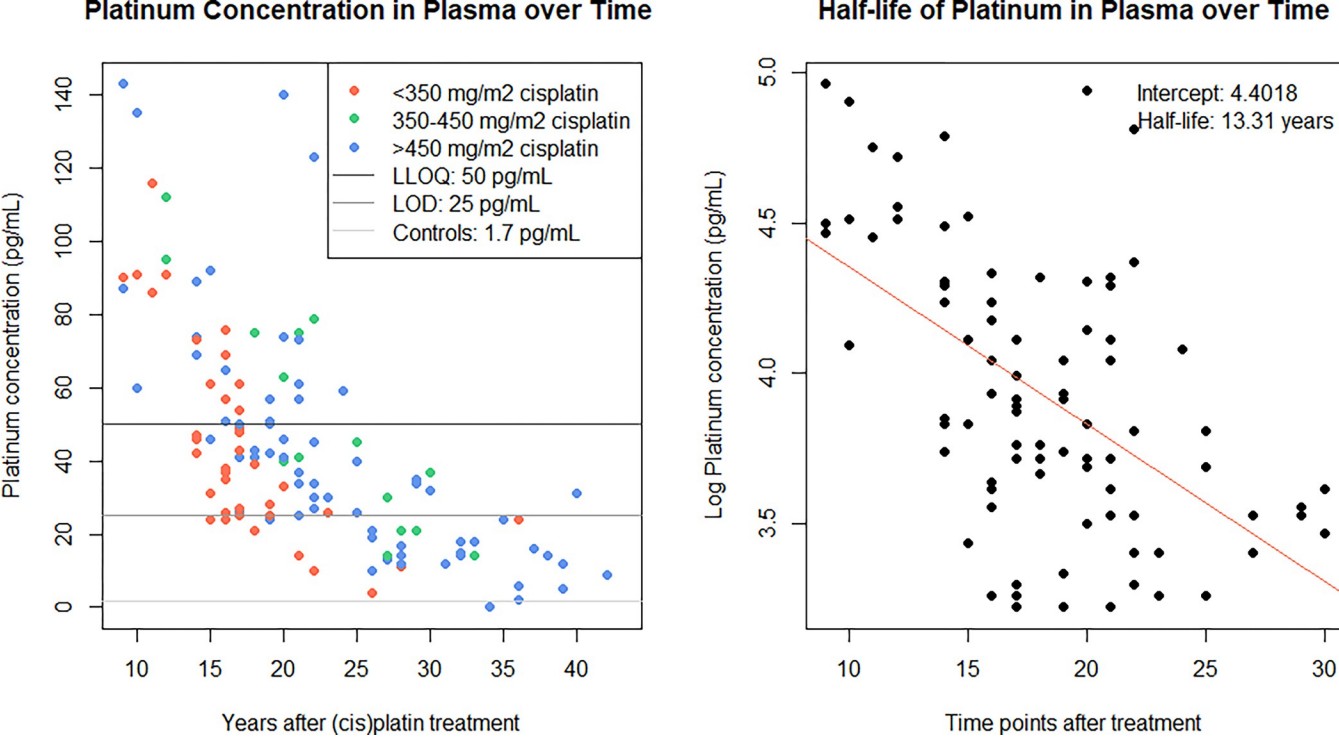

**Fig 1.** A&B. Correlation between platinum concentration in plasma and years after cisplatin treatment and half-life of platinum in plasma.

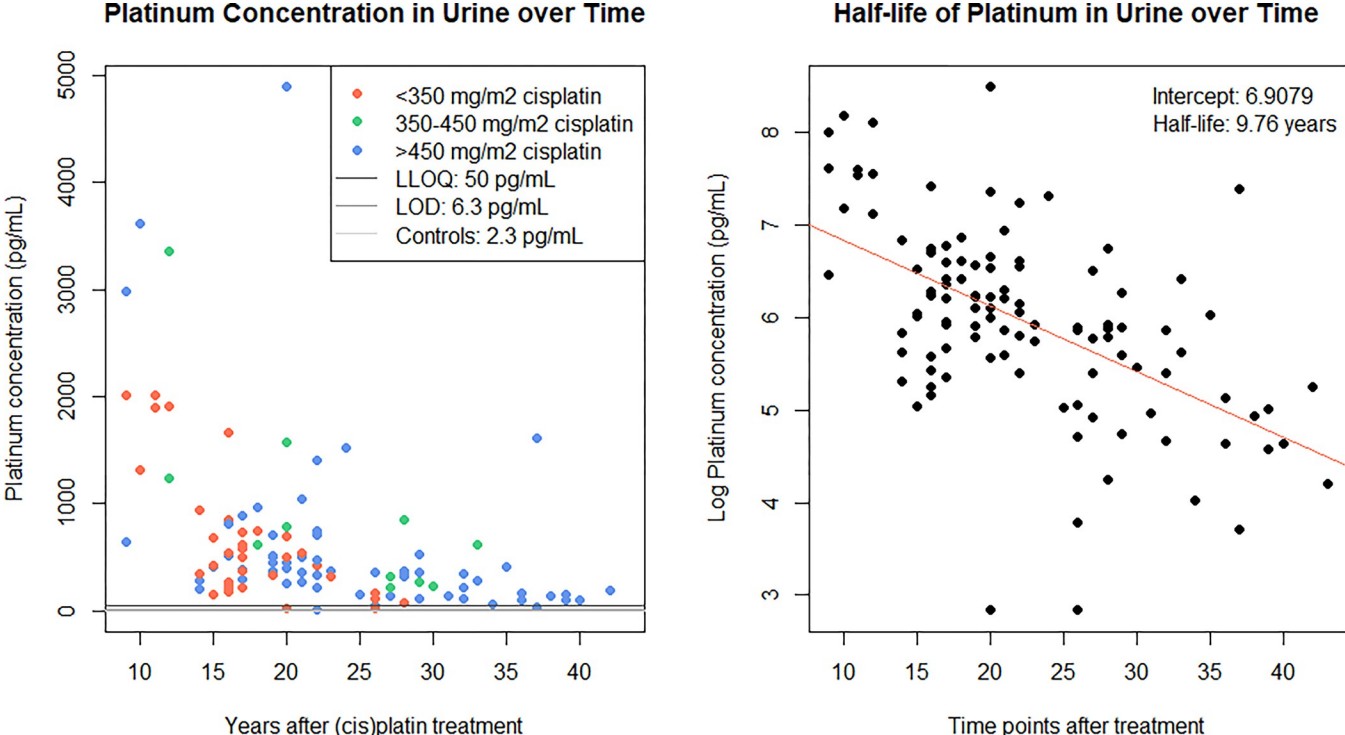

**Fig 2.** A&B. Correlation between platinum concentration in urine and years after cisplatin treatment and half-life of platinum in urine.

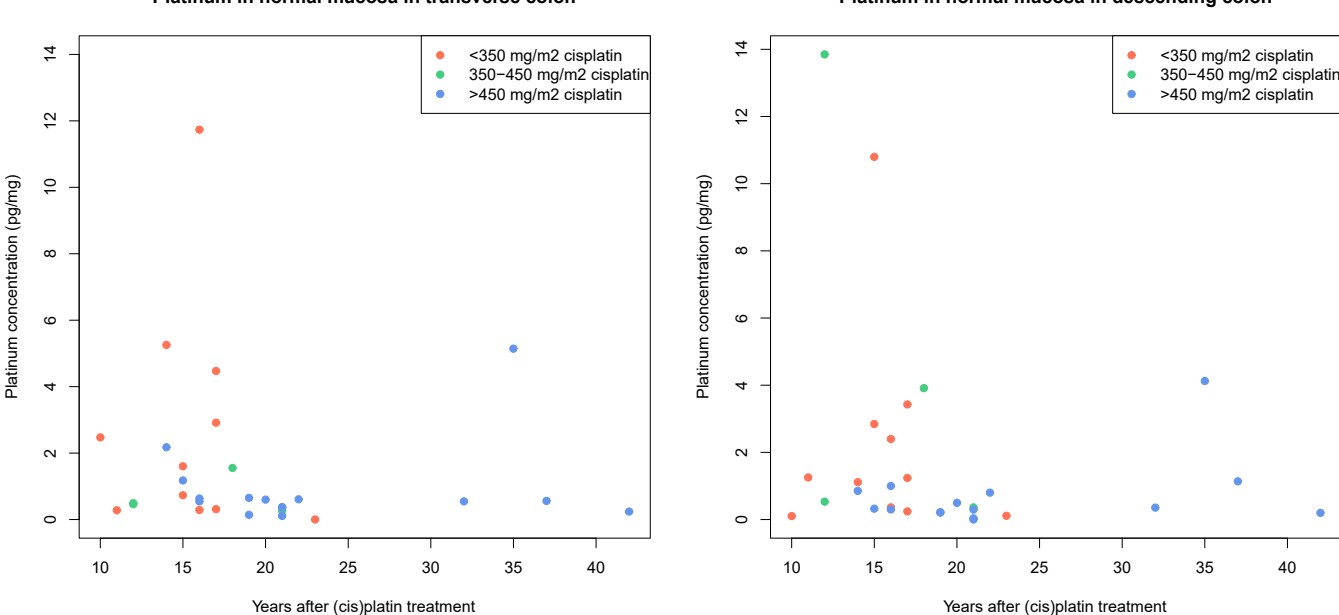

**Fig 3.** A&B. Correlation between platinum concentration in the mucosa of the transverse and descending colon and years after cisplatin treatment.

## Platinum in normal colonic mucosa

The median platinum concentrations were similar in the transverse colon (n = 30, 0.58 pg/mg [IQR: 0.33–1.59]; Fig 3A) and in the descending colon (n = 30, 0.51 pg/mL [IQR 0.26–1.25]; p = 0.62; Fig 3B). A total of 55% of platinum concentrations in normal colonic mucosa of TCS was above the LLOQ (10 pg/mL), 40% was between the LLOQ and the LOD (2 pg/mL); and 5% was below the LOD and not used for the calculations. The LLOQ concentrations are dependent on the weight of the biopsies. Based on a mean biopsy weight of 3.5 mg of study participants, the LLOQ and LOD were 0.5 pg/mg and 0.1 pg/mg, respectively. All platinum concentrations in control samples (n = 9) were below the LOD. There was no correlation between the number of cycles of cisplatin and the platinum concentration in normal colonic mucosa; 20–29 years after treatment, the platinum concentration was significantly lower compared to 10–19 years after treatment (S3 Table in S1 File).

## Correlation platinum concentrations in different samples and clinical findings

Platinum concentrations tended to be higher in individuals with a shorter interval between cisplatin treatment and study enrollment (Figs 1A, 2A and 3A, 3B). There was a statistically significant correlation between plasma and urine platinum concentrations (Spearman correlation coefficient r = 0.67(95%CI 0.54–0.78, p<0.001). In 29 samples in which platinum concentrations could be determined for both plasma, urine and normal colonic mucosa, the correlation between platinum concentrations in plasma and urine was less clear (r = 0.69) and not statistically significant (p = 0.43; S1 Fig). There was relatively poor correlation between platinum concentrations in plasma or urine and the platinum concentration in the normal colonic mucosa samples (plasma-colon: r = 0.13 (p = 0.28), urine-colon: r = 0.04 (p = 0.15); S1 Fig). The median platinum concentration did not increase in any of the samples when no adenomas, non-advanced adenomas/SPs, or advanced neoplasia were detected during

colonoscopy. Median platinum concentrations in plasma (p = 0.7), transverse colon (p = 0.06), and descending colon (p = 0.53) did not differ between individuals with no lesions, non-advanced adenomas/SPs or AN as their most advanced lesion detected at colonoscopy (S2 Fig).

## Discussion

In this study we found measurable platinum concentrations up to 40 years after treatment, in plasma, urine, and normal colonic mucosa samples from TCS treated with cisplatin. Platinum concentrations were higher in all three different types of patient samples compared to control samples.

The measured platinum concentrations decreased with time since cisplatin treatment. Several other studies have shown long-term retention of platinum in plasma and urine [6–8, 11, 12]. However, no study has shown that platinum persists in these tissues beyond 20 years after treatment and has evaluated correlations between platinum concentrations in plasma, urine and normal colonic mucosa. To our knowledge, this is the first study which quantified platinum concentrations in normal colonic mucosa of patients exposed to cisplatin during cancer treatment. Almost all normal colonic mucosa of cisplatin-treated TCS contained higher platinum concentrations than controls (i.e., measurable platinum above the LOD), suggesting that platinum may not only be measurable for a very long time in plasma and urine, but might also be retained in various tissues of the human body.

A recent epidemiologic cohort study showed a higher risk of developing CRC in TCS treated with platinum-based chemotherapy compared to TCS not treated with platinum-based chemotherapy (HR: 3.9) [4]. Based on these findings, we evaluated platinum concentrations in normal colonic mucosa in TCS and correlated them with the most advanced lesion detected by colonoscopy. Although we did not find a correlation between platinum concentrations in colonic mucosa and clinical outcomes (i.e., AN or CRC development), it has been hypothesized that long-term accumulation of platinum in (healthy) tissues may be associated with early ageing through cellular senescence [13]. However, cisplatin leads to DNA damage, which could also occur in healthy tissue at the time of cisplatin treatment, therefore increasing the risk of developing cancer in TCS and shifting the cancer risk to a younger age. This is supported by a recent study showing that oxaliplatin treatment leads to increased mutational load in stem cells of normal colonic mucosa [14]. The long-term retention of platinum in plasma is likely due to the slow release of platinum from regenerating tissues throughout the human body. Brouwers et al. found that long after treatment, platinum in plasma still had remaining protein binding capacity, implicating that even years after treatment, around 10% of circulating platinum may still be reactive in patients. Furthermore, given the extensive binding of cisplatin to proteins, it is to be expected that platinum is gradually released into the bloodstream when tissues regenerate, after which renal excretion is initiated [7, 8].

In a study conducted by Hjelle et al., it was demonstrated that out of 76 TCS, of whom 12 developed an SMN, a lower risk of SMNs was associated with more rapid decreases in plasma platinum levels [15]. Taken together, we hypothesize that the long-term presence/retention of active platinum among TCS treated years before with cisplatin may contribute to the accumulation of somatic mutations in normal tissues, which might enhance the mutations that developed during the cisplatin treatment. This treatment-related accumulation of DNA mutations then adds to age-related accumulation of somatic mutations caused by endogenous mutagenic processes, thus leading to higher risk of developing SMNs. The emergence of other fourth-generation platinum agents, which appear to show a similar mechanism of action but a reduced carcinogenic effect on non-malignant cells in vitro and in vivo, promises lower rates of late side effects in the future [16].

The major strength of our study was our ability to assess platinum concentrations in different types of patient samples from TCS in a well-defined cohort up to 40 years after initial treatment using ICP-MS, a highly sensitive technique for quantification in different samples. In addition, we were able to assess correlations between measurements in the different samples and colonoscopy outcomes in relation to platinum measurements in plasma, urine, and normal colonic mucosa. Further research is needed to determine the relationship between platinum exposure and the subsequent development of CRC in normal colonic mucosa.

Almost all urine samples had platinum concentrations above the LLOQ, whereas this has not occured for plasma samples, although most platinum concentrations in plasma were well above the LOD and higher than those in unexposed controls. The inability to distinguish between unchanged cisplatin and its metabolites or adducts limits the use of ICP-MS to measure platinum in biological samples. As a result, information on the composition of platinum components was not obtained. The renal function at time of treatment could also have influenced long-term platinum concentrations in biological samples. Unfortunately, no data were available on the glomerular filtration rate at time of cisplatin treatment, although none of the study participants suffered from renal insufficiency at the time of sample acquisition. A previous study also showed that plasma and urinary platinum concentrations were strongly correlated years after cisplatin treatment, which was confirmed by the high correlation between platinum in plasma and urine in our study [7]. These observations suggest that the effect of renal excretion on plasma platinum concentrations at follow-up is minimal, which was underlined by the fact that the platinum half-life in plasma was comparable to that in urine.

In conclusion, the use of cisplatin can result in long-term exposure to low doses of circulating platinum and platinum accumulation/retention in various types of patient samples, and may be associated with an increased risk of cancer through induction of somatic mutations and thereby partly explain the increased SMN risk in TCS. Individuals exposed to cisplatin should be carefully monitored because of the potential long-term effects of platinum accumulation, and fourth-generation platinum agents may offer future solutions to alleviate risk of these late effects.

## Supporting information

**S1 File. Association between number of cycles of cisplatin and years since treatment and the log-transformed platinum concentration in plasma, urine and normal colonic mucosa.** (DOCX)

**S1 Fig. Correlation between platinum concentrations in plasma, urine, and normal colon tissue of 29 patients.** (TIF)

**S2 Fig. Median platinum concentrations in plasma, transverse and descending colon according to findings at colonoscopy.** (TIFF)

## Acknowledgments

We would like to thank all CATCHER study participants. We thank the Bioanalytic Laboratory of the Netherlands Cancer Institute for preparation of samples and performing the ICP-MS analyses.

We would also like to thank the CATCHER study working group: Tanya M. Bisseling, Leon M. G. Moons, Manon C. W. Spaander, Inge Huibregtse, Dorien van der Biessen-van Beek, Sasja F. Mulder, Lisette Saveur, Martijn Kerst, Danielle Zweers, Britt B. M. Suelmann, Ronald

de Wit, Agnes Reijm, Sophia van Baalen, Anneke J. van Vuuren, Hendrik Messal, Guillaume Belthier, and Jacco van Rheenen.

## Author Contributions

**Conceptualization:** Emilie C. H. Breekveldt, Berbel L. M. Ykema, Alwin D. R. Huitema, Jos H. Beijnen, Petur Snaebjornsson, Michael Schaapveld, Flora E. van Leeuwen, Hilde Rosing, Monique E. van Leerdam.

**Data curation:** Emilie C. H. Breekveldt, Michael Schaapveld, Hilde Rosing.

**Formal analysis:** Emilie C. H. Breekveldt, Jos H. Beijnen, Michael Schaapveld, Hilde Rosing.

**Investigation:** Emilie C. H. Breekveldt.

**Methodology:** Emilie C. H. Breekveldt, Alwin D. R. Huitema, Jos H. Beijnen, Petur Snaebjornsson, Michael Schaapveld, Flora E. van Leeuwen, Hilde Rosing, Monique E. van Leerdam.

**Project administration:** Emilie C. H. Breekveldt, Berbel L. M. Ykema, Petur Snaebjornsson, Hilde Rosing.

**Supervision:** Michael Schaapveld, Monique E. van Leerdam.

**Validation:** Emilie C. H. Breekveldt, Hilde Rosing.

**Visualization:** Emilie C. H. Breekveldt.

**Writing – original draft:** Emilie C. H. Breekveldt.

**Writing – review & editing:** Emilie C. H. Breekveldt, Berbel L. M. Ykema, Alwin D. R. Huitema, Jourik A. Gietema, Jos H. Beijnen, Petur Snaebjornsson, Michael Schaapveld, Flora E. van Leeuwen, Hilde Rosing, Monique E. van Leerdam.

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
