## [Decision Letter · Decision Letter 0]

24 Mar 2024

PONE-D-24-02752Platinum Retention in Plasma, Urine, and Normal Colonic Mucosa in Cisplatin-Treated Testicular Cancer SurvivorsPLOS ONE

Dear Dr. Breekveldt,

Thank you for submitting your manuscript to PLOS ONE. After careful consideration, we feel that it has merit but does not fully meet PLOS ONE’s publication criteria as it currently stands. Therefore, we invite you to submit a revised version of the manuscript that addresses the points raised during the review process.

We look forward to receiving your revised manuscript.

Kind regards,

Niraj Babu

Academic Editor

PLOS ONE

Journal Requirements:

P. Snaebjornsson has done unrelated consultancy for MSD and Bayer and received payment from MEDtalks for educational presentation. All other authors declare no conflicts of interest.

We note that one or more of the authors are employed by a commercial company.

“The funder provided support in the form of salaries for authors, but did not have any additional role in the study design, data collection and analysis, decision to publish, or preparation of the manuscript. The specific roles of these authors are articulated in the ‘author contributions’ section.”

3. We are unable to open your Supporting Information file Supp Fig 2.eps. Please kindly revise as necessary and re-upload.

Additional Editor Comments:

Dear authors,

Thank you for submitting the manuscript to PLOS One. The manuscript has been reviewed by two potential reviewers who have unequivocally expressed interest in the manuscript and its results. The authors are recommended to submit a revised manuscript or explanation for the comments presented by the peer reviewers.

Reviewers' comments:

Reviewer's Responses to Questions

**Comments to the Author**

1. Is the manuscript technically sound, and do the data support the conclusions?

Reviewer #1: Yes

Reviewer #2: Yes

2. Has the statistical analysis been performed appropriately and rigorously? 

Reviewer #1: Yes

Reviewer #2: Yes

3. Have the authors made all data underlying the findings in their manuscript fully available?

Reviewer #1: Yes

Reviewer #2: Yes

4. Is the manuscript presented in an intelligible fashion and written in standard English?

Reviewer #1: Yes

Reviewer #2: Yes

5. Review Comments to the Author

Reviewer #1: This is a study of testicular germ cell tumor survivors where they captures plasma and urine samples from survivors as well as colonic biopsies done at time of colonoscopy. They showed that there is still detectable platinum in sample up to 40 years post therapy (and had controls who had undetectable levels). They found levels of platinum in normal colonic mucosa of the transverse and descending colon.

GENERAL COMMENTS

Interesting study that re-confirms previous finding that platinum agents remain in the body 20 years post therapy. This adds to this literature with data going out 40 years post therapy, as well as showing levels in colon samples done at time of colonoscopy. The postulate that this could result in future second malignancies in survivors.

Very well written paper - I have very little to suggest.

Very publishable.

SPECIFIC COMMENTS

Abstract:

1. May wish to put the total sample size of “154 individuals” in the abstract methods section

2. Consider in abstract methods putting “(n=2 per 30 randomly selected patient undergoing colonoscopy)”

Introduction:

No comments - well written

Methods:

No comments - clear.

Results:

1. Not sure if you have this data - but what dosing regimen of cisplatin was used for these patients. I am assuming that a dose of 20 mg/m2 per day for 5 days was used (as this is the typical North American dosing for testicular cancer) - can you confirm what dosing was used in the Netherlands by the participants? Did all receive the same dosing - and if not did this impact the results.

2. Any difference in those who received 3 vs 4 doses of cisplatin (did those with higher cumulative exposure have higher levels)?

Discussion:

Well written and agree with the conclusions.

Reviewer #2: The primary outcomes of this CATCHER study, a colonoscopy screening study, are the total platinum concentrations in plasma, urine, and normal colonic mucosa samples of the transverse and descending colon. Secondary outcomes include platinum half-lives in plasma and urine, as well as median platinum concentrations by the most advanced lesion at colonoscopy.

Overall, the statistical considerations are presented comprehensively; however, there are several minor statistical concerns about this manuscript.

Statistical critiques:

1. There is no sample size justification for this study. The authors should conduct a precision analysis to evaluate the precision of the analysis results.

2. The authors should provide the 95% confidence intervals (CIs) for all major findings, e.g., the half-life of platinum in plasma over time.

3. The authors may consider using a cubic spline method instead of linear regression.

4. Please use the Spearman correlation coefficient to evaluate the correlations between plasma and urine concentrations of platinum.

5. The authors may conduct a multivariable data analysis to evaluate the platinum concentration in plasma/urine over time adjusted for possible confounding variables.

6. It is unclear whether any study subjects experienced a concentration below the limit of detection, as all controls were below the limit of detection.

6. PLOS authors have the option to publish the peer review history of their article (what does this mean?). If published, this will include your full peer review and any attached files.

Reviewer #1: **Yes: **S. Rod Rassekh

Reviewer #2: No

---

## [Author Response · Author response to Decision Letter 0]

12 Jul 2024

Dear Reviewers, 

Please find attached our rebuttal letter and responses to your valuable comments. 

Best regards, 

Emilie Breekveldt

---

## [Decision Letter · Decision Letter 1]

17 Oct 2024

Platinum Retention in Plasma, Urine, and Normal Colonic Mucosa in Cisplatin-Treated Testicular Cancer Survivors

PONE-D-24-02752R1

Dear Dr. Breekveldt,

We’re pleased to inform you that your manuscript has been judged scientifically suitable for publication and will be formally accepted for publication once it meets all outstanding technical requirements.

Kind regards,

Niraj Babu

Academic Editor

PLOS ONE

Additional Editor Comments (optional):

Reviewers' comments:

Reviewer's Responses to Questions

**Comments to the Author**

1. If the authors have adequately addressed your comments raised in a previous round of review and you feel that this manuscript is now acceptable for publication, you may indicate that here to bypass the “Comments to the Author” section, enter your conflict of interest statement in the “Confidential to Editor” section, and submit your "Accept" recommendation.

Reviewer #1: All comments have been addressed

Reviewer #2: All comments have been addressed

2. Is the manuscript technically sound, and do the data support the conclusions?

Reviewer #1: Yes

Reviewer #2: Yes

3. Has the statistical analysis been performed appropriately and rigorously? 

Reviewer #1: Yes

Reviewer #2: Yes

4. Have the authors made all data underlying the findings in their manuscript fully available?

Reviewer #1: Yes

Reviewer #2: Yes

5. Is the manuscript presented in an intelligible fashion and written in standard English?

Reviewer #1: Yes

Reviewer #2: Yes

6. Review Comments to the Author

Reviewer #1: The authors have addressed the comments provided to them and the paper is well written and read for publication.

Reviewer #2: The authors have responded well to the statistical issues raised in the previous review. There is no further statistical concern about this revised manuscript.

7. PLOS authors have the option to publish the peer review history of their article (what does this mean?). If published, this will include your full peer review and any attached files.

Reviewer #1: **Yes: **Shahrad Rod Rassekh

Reviewer #2: No

---

## [Editor Report · Acceptance letter]

5 Nov 2024

PONE-D-24-02752R1 

PLOS ONE

Dear Dr. Breekveldt, 

I'm pleased to inform you that your manuscript has been deemed suitable for publication in PLOS ONE. Congratulations! Your manuscript is now being handed over to our production team.

Kind regards, 

on behalf of

Dr. Niraj Babu 

Academic Editor

PLOS ONE